# Natural selection favoring more transmissible HIV detected in United States molecular transmission network

Joel O. Wertheim[1]*, Alexandra M. Oster[2], William M. Switzer [2], Chenhua Zhang[3,4], Nivedha Panneer[2], Ellsworth Campbell[2], Neeraja Saduvala[3], Jeffrey A. Johnson[2] & Walid Heneine[2]

HIV molecular epidemiology can identify clusters of individuals with elevated rates of HIV transmission. These variable transmission rates are primarily driven by host risk behavior; however, the effect of viral traits on variable transmission rates is poorly understood. Viral load, the concentration of HIV in blood, is a heritable viral trait that influences HIV infectiousness and disease progression. Here, we reconstruct HIV genetic transmission clusters using data from the United States National HIV Surveillance System and report that viruses in clusters, inferred to be frequently transmitted, have higher viral loads at diagnosis. Further, viral load is higher in people in larger clusters and with increased network connectivity, suggesting that HIV in the United States is experiencing natural selection to be more infectious and virulent. We also observe a concurrent increase in viral load at diagnosis over the last decade. This evolutionary trajectory may be slowed by prevention strategies prioritized toward rapidly growing transmission clusters.

[1] Department of Medicine, University of California, San Diego, CA, USA. [2] Division of HIV/AIDS Prevention, Centers for Disease Control and Prevention, Atlanta, GA, USA. [3] ICF International, Atlanta, GA, USA. [4] Present address: SciMetrika LLC, Atlanta, GA, USA. *email: jwertheim@ucsd.edu

Natural selection is the process by which the differential reproductive success of an organism with particular trait, whose variance in the population has a genetic under-pinning (i.e., is heritable), leads to change in a population. In human immunodeficiency virus (HIV), a trait that is likely shaped by natural selection is viral load, the concentration of HIV in blood[1,2]. Viral load is an established proxy of HIV sexual infectiousness: the probability of viral transmission per sexual event[3–6]. Set-point viral load (SPVL), the stable viral load during the asymptomatic stage of HIV infection[7], is a heritable viral trait in antiretroviral therapy (ART)-naïve individuals that influences both HIV transmission and rate of disease progression[8–18].

Measuring natural selection associated with SPVL is not straightforward, because higher viral loads are also associated with higher infectiousness[6,19,20]. Higher SPVLs in untreated persons result in higher infectiousness and shorter progression times to acquired immunodeficiency syndrome (AIDS)[21,22]. A $\log_{10}$ increase in viral load is associated with a 100% increase in per-event HIV transmission risk, and even smaller increments of 0.3 $\log_{10}$ and 0.5 $\log_{10}$ increase per-event transmission risk by 20% and 40%, respectively, underscoring the potential impact of viral load on HIV spread at the population level[20]. Importantly, higher infectiousness due to increased viral load may not necessarily result in more transmission at the population level, because higher SPVL comes with an associated increase in the rate of disease progression that limits the duration of time over which transmission can occur[1,2]. To measure the strength and direction of natural selection on viral load, an approach must capture differential reproductive success of viral variants across multiple individuals over time.

Clustering in an HIV-1 molecular transmission network can serve as a proxy for transmission rate of the virus across multiple individuals and, thus, for efficiency of spread in a population[23–26]. The primary drivers of this population-level variability in HIV transmission rates are host transmission risk behavior, host demography, and the underlying connectivity of partner networks[23,27,28]. Nonetheless, recent advances in HIV surveillance facilitated by the availability of viral sequences from antiviral drug resistance testing has allowed the analysis of large national sequence databases and the concomitant ability to identify viral traits that may impact HIV spread[23,24,27,28]. In recent work with the United States National HIV Surveillance System (NHSS) database, we have used the frequency of viral genetic clustering in a molecular transmission network as a proxy for viral transmission fitness, or the relative rate of spread of a given viral genotype across the population. Using this approach, we were able to assess how drug resistance-associated mutations (DRAMs) alter transmission fitness[29]. We found that HIV strains containing the DRAM M184V significantly reduced genetic clustering compared with wild-type HIV, reflecting a stark decrease in transmission fitness. In contrast, other DRAMs, such as K103N or L90M, had transmission fitness that was similar to or exceeded wild-type, permitting the establishment of large self-sustaining reservoirs of drug-resistant virus. In support of our approach, a highly para-meterized phylodynamic analysis estimating the transmission fitness of strains containing DRAMs in the Swiss HIV cohort reached identical conclusions[30].

Having shown that the relative frequency of genetic clustering in a large transmission network successfully approximated the transmission fitness of HIV containing DRAMs[29,30], we posit that the same approach can also be used to identify and characterize differences in transmission fitness of cocirculating wild-type HIV, by examining the frequency and intensity of clustering of HIV with different SPVLs. Here, we employ this molecular epidemiological approach to answer the question of whether circulating wild-type strains of HIV in the United States differ in their transmission fitness and whether frequently transmitted viruses (those in genetic clusters) are more infectious than less frequently transmitted (nonclustered) viruses.

In this study, we analyze viral load data as a marker of infectiousness and genetic clustering as a marker of the relative transmission fitness from >40,000 well-characterized ART-naïve individuals, with an HIV diagnosis in the United States NHSS database. We report robust evidence that frequently transmitted strains, which are found in genetic transmission clusters, have significantly higher viral loads than nonclustered viruses. This finding, combined with an associated increase in viral load at diagnosis over the past decade, suggests that circulating HIV strains in the United States are under natural selection favoring higher infectiousness. We discuss the implications on HIV prevention efforts targeted to interrupt transmission clusters and on the broader evolutionary trajectory of HIV infectiousness and virulence.

## Results

**Molecular transmission network.** Of the 251,754 individuals in the NHSS database with a reported HIV-1 polymerase (*pol*) sequence, 41,409 were ART-naïve at diagnosis and had a reported HIV-1 subtype B resistance genotype performed within three months of diagnosis (see Methods for detailed inclusion criteria). Using *pol* sequences from these 41,409 individuals (31,285 of whom had wild-type virus containing no DRAMs), we inferred a total of 4366 molecular transmission clusters using a genetic distance threshold of ≤0.015 substitutions/site in HIV-TRACE (HIV TRAnsmission Cluster Engine)[31], comprising 17,688 persons (42.7%). Of the 33,285 individuals with wild-type virus, 24,028 (72.2%) had a reported viral load measurement taken three months prior to or one-month post genotyping and 9015 (37.5%) of these individuals were genetically linked to another wild-type virus in this network (Table 1). As expected for individuals with recent infection[32], a higher frequency of clustering was observed for individuals with HIV diagnosed in earlier stages of infection, highlighting the importance of stratifying our analyses by stage of infection (Table 1).

**Viral load across the transmission network.** We detected a robust association between viral load and clustering in the inferred molecular transmission network (Table 2; Fig. 1). Infections diagnosed during stages 1, 2, and 3 had significantly higher first viral load measurement, if they were clustered in the network (Fig. 2a). For individuals with HIV diagnosed during stage 1 infection, the first viral load measurement was used as a proxy for SPVL. The median SPVL in clustered individuals was 0.110 $\log_{10}$ copies/ml higher than in nonclustered individuals, after adjusting for epidemiologic and laboratory covariates (multivariate linear regression; $p < 0.001$). Clustered individuals with HIV diagnosed during stage 2 and stage 3 had 0.107 $\log_{10}$ and 0.050 $\log_{10}$ copies/ml higher viral load than nonclustered individuals, respectively (multivariate linear regression; $p < 0.001$ and $p = 0.010$). There was no significant difference in viral loads in clustered versus nonclustered individuals with HIV diagnosed during stage 0 (multivariate linear regression; $p = 0.496$).

To provide a rough approximation of the impact of higher viral load on transmission fitness in our dataset, we inverted the regression analysis to estimate the effect of viral load on clustering. For infections diagnosed during stage 1, a one $\log_{10}$ increase in SPVL increased the adjusted odds of clustering by 1.12: a 12% increase in relative fitness advantage (multivariate logistic regression; $p < 0.001$).

**Viral load over time.** We examined temporal trends in first viral load postdiagnosis from all ART-naïve individuals (i.e., both

**Table 1 Median viral load (VL) in clustered and nonclustered individuals at different stages of HIV infection at diagnosis in people with wild-type virus, United States.**

| Stage | # Cases | Median VL[a] | Clustered | | Nonclustered | | ΔLog$_{10}$ VL[b] |
|---|---|---|---|---|---|---|---|
| | | | # Cases (%) | Median VL[a] | # Cases (%) | Median VL[a] | |
| All | 24,028 | 48,966 | 9015 (37.5%) | 45,107 | 15,013 (62.5%) | 51,286 | −0.056 |
| 0 | 476 | 78,093 | 257 (54.0%) | 81,730 | 219 (46.0%) | 74,580 | 0.040 |
| 1 | 5914 | 18,700 | 2787 (47.1%) | 22,517 | 3127 (52.9%) | 16,330 | 0.140 |
| 2 | 9337 | 38,470 | 3991 (42.7%) | 46,266 | 5346 (57.3%) | 33,423 | 0.141 |
| 3 | 7280 | 122,000 | 1528 (21.0%) | 144,619 | 5752 (79.0%) | 119,031 | 0.085 |
| Unknown[c] | 1021 | 33,200 | 452 (44.3%) | 37,250 | 569 (55.7%) | 30,125 | 0.092 |

[a]First reported VL (copies/ml) three months prior to or one-month post genotyping
[b]Difference in median log$_{10}$ VL between clustered and nonclustered cases
[c]Individuals with an indeterminate stage of diagnosis[75]

**Table 2 Relationship between attributes and viral load in the multivariate linear regression analysis for individuals with wild-type virus in the inferred molecular transmission network, stratified by stage of infection at diagnosis, United States.**

| Variable | Attribute | Adjusted beta/significance | | | |
|---|---|---|---|---|---|
| | | Stage 0 | Stage 1 | Stage 2 | Stage 3 |
| Clustered | Yes | 0.053 | 0.110*** | 0.107*** | 0.050* |
| | No | Ref | Ref | Ref | Ref |
| Birth sex | Male | 0.230 | 0.180*** | 0.160*** | 0.025 |
| | Female | Ref | Ref | Ref | Ref |
| Transmission risk factor | Male–male sexual contact | Ref | Ref | Ref | Ref |
| | Unknown/other | 0.216 | −0.073* | −0.094*** | −0.016 |
| | Heterosexual contact | −0.045 | −0.156*** | −0.103*** | −0.042 |
| | Injection drug use | 0.625 | −0.013 | −0.035 | −0.073 |
| | Male–male sexual contact and injection drug use | 0.217 | 0.073 | 0.045 | 0.051 |
| Race/ethnicity | Black/African American | Ref | Ref | Ref | Ref |
| | Hispanic/Latino | 0.135 | 0.100*** | 0.122*** | 0.097*** |
| | Other | −0.019 | 0.109* | 0.076* | 0.089* |
| | White | 0.108 | 0.156*** | 0.183*** | 0.126*** |
| Diagnosis age (years) | 13–19 | −0.263 | 0.024 | 0.117*** | 0.045 |
| | 20–29 | Ref | Ref | Ref | Ref |
| | 30–39 | 0.017 | 0.036 | 0.029 | 0.027 |
| | 40–49 | 0.215 | 0.154*** | 0.090*** | 0.065* |
| | 50–59 | 0.031 | 0.141*** | 0.094*** | 0.059* |
| | 60+ | 0.021 | 0.272*** | 0.232*** | 0.058 |
| Δ 100 CD4$^+$ count[a] | — | −0.063*** | −0.002 | −0.094*** | −0.249*** |
| Diagnosis year | — | 0.010 | 0.016*** | 0.010*** | 0.008** |

***p < 0.001; **p < 0.01; *p < 0.05
[a]Increase of 100 CD4$^+$ cells/mm$^3$

clustered and nonclustered) with a reported subtype B genotype and no evidence of DRAMs. These viral loads increased significantly over the time period analyzed for infections diagnosed during stages 1, 2, and 3 (univariate regression; $p < 0.001$; Fig. 3). For individuals with stage 1 infection at diagnosis, viral load has increased an average of 0.016 log$_{10}$ copies/ml per year. In 2007, median SPVL at diagnosis was 13,020 copies/ml, and by 2016 it was 22,100 copies/ml. Over the period of a decade, SPVL increased by over 0.2 log$_{10}$ copies/ml in this population. Similar patterns were seen for individuals diagnosed with stage 2 and stage 3 infection. This association between viral load and year of diagnosis was robust in the univariate and multivariate regression models (Table 2; Fig. 3). We detected no such association for infections diagnosed at stage 0, possibly owing to the fewer number of cases, shorter time frame of reporting, and the rapidly shifting dynamics of viral load during acute infection[33].

The frequency of clustering also increased over the time period analyzed (Supplementary Fig. 1), although it has been relatively stable since 2009. Nonetheless, the inferred association between clustering and viral load was robust to the inclusion of year of diagnosis as a covariate in the regression model. This association was also robust to the inclusion of demographic and transmission risk factor covariates (Table 2), which are consistently found to be major factors of variation in transmission rate across molecular transmission networks[23–26].

We found no evidence of a significant interaction between clustering and year of diagnosis on viral load in the multivariate regression model, at any stage of infection at diagnosis. Therefore, the rate of increase in mean viral load over time was not significantly different in clustered and nonclustered individuals. If natural selection is acting to increase viral load, the strength of this selection has not changed over the time period analyzed. Moreover, this increase in viral load over time is occurring across the entire sampled HIV population, rather than only in clustered viruses.

**Progressive effect of network connectivity on viral load.** The more connected an individual was in the network (i.e., increase in

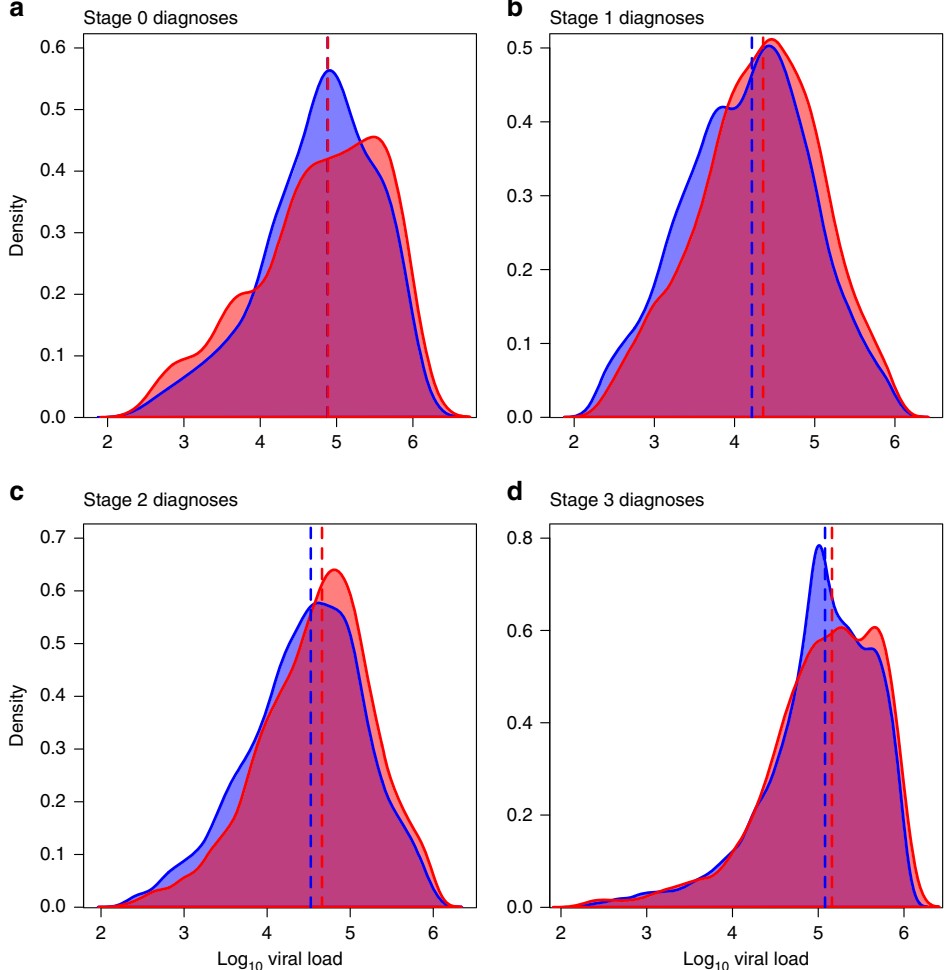

**Fig. 1 Density distributions for log$_{10}$ viral load at diagnosis by network clustering.** Viral loads (copies/ml) for individuals with **a** stage 0, **b** stage 1, **c** stage 2, and **d** stage 3 infection at diagnosis are displayed separately. Viral loads (copies/ml) from individuals who clustered in the network shown in red; individuals who are not clustered in the network shown in blue; overlap shown in purple. Median values are depicted as dashed lines.

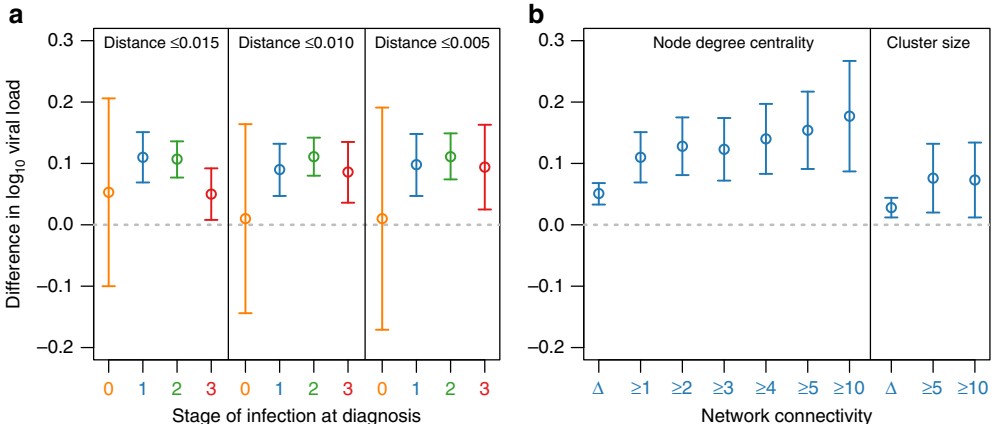

**Fig. 2 Increase in viral load for clustered versus nonclustered individuals. a** Betas from the multivariate regression model for the difference in log$_{10}$ viral load (copies/ml) in clustered individuals versus nonclustered at different stages of infection at diagnosis and different genetic distance thresholds: ≤0.015 substitutions/site, ≤0.010 substitutions/site, and ≤0.005 substitutions/site. Stage of infection is denoted by color. **b** Betas from the multivariate regression model for difference in log$_{10}$ viral load for individuals diagnosed at stage 1 infection with increasing node degree centrality (i.e., number of genetically linked partners) and cluster size relative to nonclustered individuals. Error bars represent the 95% confidence intervals for these estimates of beta. Node degree centrality compares individuals with at least that degree versus nonclustered individuals. Hence, node degree ≥1 in **b** is equivalent to clustered versus nonclustered depicted in **a**. Cluster size compares individuals in clusters of at least that size versus individuals in clusters of small sizes (i.e., cluster size ≥5 versus cluster size <5). Δ denotes the difference in log$_{10}$ viral load for each increase in node degree centrality or cluster size. Network constructed at ≤0.015 substitutions/site. Sample sizes (*n*) for statistical tests are provided in Table 1.

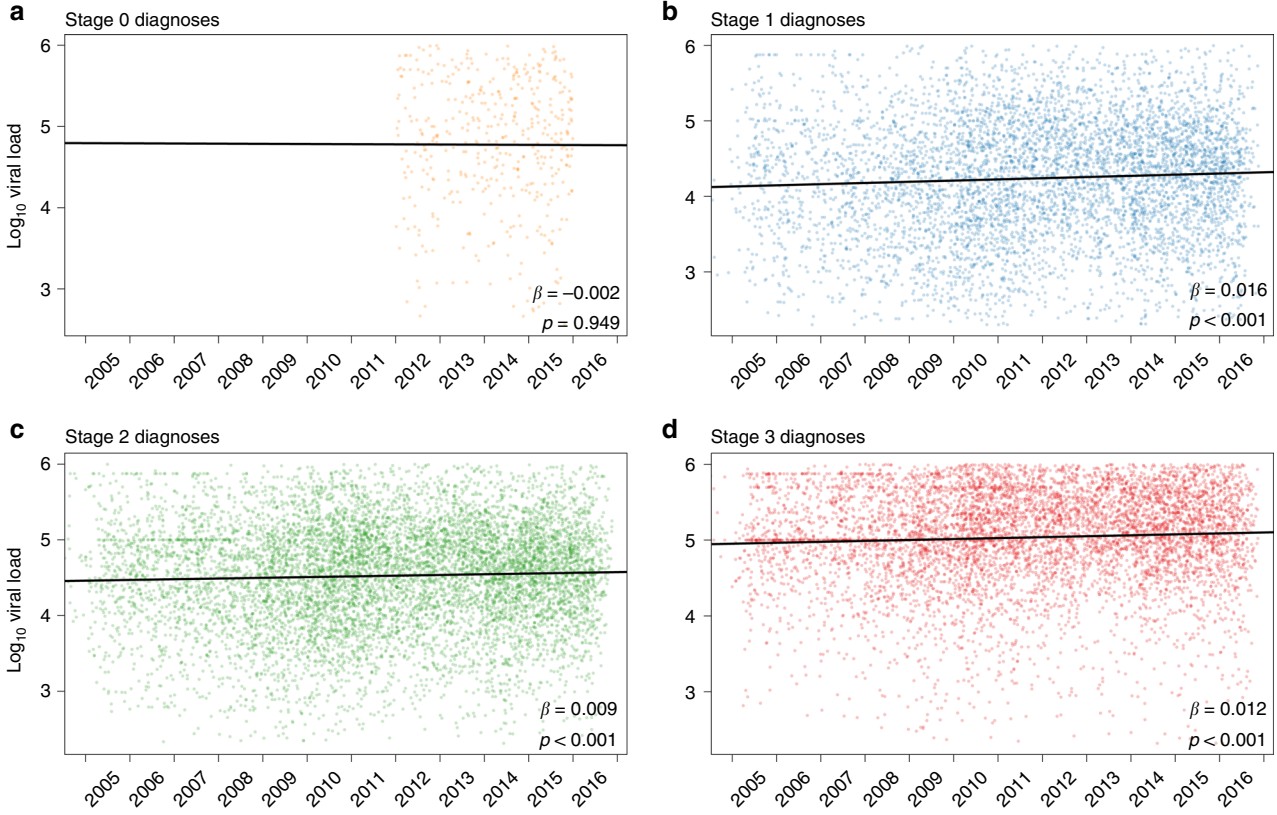

**Fig. 3 Viral load at diagnosis over time.** These plots include the first viral load measurement for individuals with **a** stage 0, **b** stage 1, **c** stage 2, and **d** stage 3 infection at diagnosis. Plots included both clustered and nonclustered individuals, all of whom were antiretroviral therapy (ART)-naive with a reported subtype B genotype and no evidence of drug resistance-associated mutations (DRAMs). Solid black lines indicate slope ($\beta$) from univariate regression analysis comparing $\log_{10}$ viral load (copies/ml) and year of diagnosis. For display purposes, viral loads for individuals with a diagnosis before 2005 are omitted, and the viral load results are plotted against their date of diagnosis, rather than only year. Sample sizes ($n$) for statistical tests are provided in Table 1.

network degree centrality), the greater the increase in viral load at HIV diagnosis. For cases diagnosed at stage 1, the addition of each additional genetic partner was associated with higher SPVL increasing 0.051 $\log_{10}$ copies/ml for each additional genetic link (Fig. 2b; multivariate linear regression; $p < 0.001$). This association was also observed when we examined this relationship restricted individuals who were already clustered (i.e., with at least one genetic partner) in the network, with a viral load increase of 0.042 $\log_{10}$ copies/ml for each additional genetic link (multivariate linear regression; $p = 0.004$). Therefore, this association is progressive across the network connectivity and not due to the difference between clustered and nonclustered individuals.

When comparing individuals with increasingly higher degree centrality (i.e., 1–10 genetically linked partners) against nonclustered individuals, the impact on SPVL is increased from 0.110 $\log_{10}$ for degree ≥1 to 0.177 $\log_{10}$ copies/ml for degree ≥10 (Fig. 2b). The same relationship between increased network connectivity and higher viral load was also detected in infections diagnosed during stage 2 (Supplementary Fig. 2).

We observed a similar relationship between increased network connectivity and higher viral load at diagnosis when examining cluster size (stage 1 in Fig. 2b; stage 2 in Supplementary Fig. 2). The addition of each member to a cluster was associated with higher SPVL for cases diagnosed at stage 1 within that cluster (multivariate linear regression; $p = 0.001$). Further, individuals in clusters with five or more members had 0.076 $\log_{10}$ copies/ml higher SPVL than individuals in clusters with fewer than five members (multivariate linear regression; $p = 0.008$). Individuals in clusters with ten or more members had a 0.073 $\log_{10}$ copies/ml

higher SPVL compared with individuals in clusters with less than ten members (multivariate linear regression; $p = 0.019$).

**Adjusting for time since infection at diagnosis**. A potential confounder when assessing the relationship between viral load and clustering in a molecular network is time between infection and diagnosis. Neglecting to stratify by diagnostic stages produces a counterintuitive result wherein the median viral load is actually higher in nonclustered individuals than clustered individuals (Table 1); however, this pattern is due to the disproportionate number of nonclustered individuals with HIV diagnosed during stage 3, who typically have viral loads an order of magnitude greater than individuals with HIV diagnosed during stage 1.

The lack of an association between viral load and clustering for individuals with HIV diagnosed at stage 0 is difficult to interpret. Viral load during the acute and early stages of HIV infection is highly dynamic, increasing and decreasing by an order of magnitude within days or weeks[33]. Further, stage 0 had the smallest sample size of any group in our analysis ($n = 476$; Table 1) and was not determined for all but a handful of cases prior to 2014 (Fig. 3; Supplementary Fig. 1), decreasing our power to detect modest effects on transmission fitness.

Within each analysis stratified by stage of infection at diagnosis, we also included CD4[+] count at the time of viral load measurement as a covariate, because CD4[+] levels will decrease as disease progresses, accounting for additional variance in viral load due to time since infection[34]. In the multivariate analysis, CD4[+]

**Table 3 Median viral load (VL) in clustered and nonclustered individuals with HIV diagnosed at stage 1 infection in people with drug-resistant virus, United States.**

| DRAM | # Cases | Median VL[a] | Clustered | | Nonclustered | | ΔLog$_{10}$ VL[b] | p value[c] |
|---|---|---|---|---|---|---|---|---|
| | | | # Cases (%) | Median VL[a] | # Cases (%) | Median VL[a] | | |
| L90M | 80 | 20,992 | 39 (48.8%) | 26,100 | 41 (51.3%) | 11,200 | 0.367 | 0.003 |
| K103N | 674 | 20,939 | 286 (42.4%) | 24,884 | 388 (57.6%) | 18,300 | 0.133 | 0.033 |
| NRTIs | 260 | 11,436 | 69 (26.5%) | 15,250 | 191 (73.5%) | 10,533 | 0.161 | 0.075 |
| Wild-type[d] | 5914 | 18,700 | 2787 (47.1%) | 22,517 | 3127 (52.9%) | 16,330 | 0.140 | <0.001 |

[a]First reported VL (copies/ml) three months prior to or one-month post genotyping
[b]Difference in median VL between clustered and nonclustered cases (not adjusted for epidemiologic and laboratory covariates)
[c]Significance of association between clustering and log$_{10}$ VL linear regression for individuals with DRAM(s) in linear regression model
[d]Wild-type virus containing no DRAMS showing same results as in shown Table 1 for individuals with HIV diagnosed during stage 1 infection

count was negatively associated with viral load for infections diagnosed during stage 0, stage 2, and stage 3 (multivariate linear regression; $p < 0.001$; Table 2). However, for infections diagnosed during stage 1, there was no association between CD4$^+$ count and viral load (multivariate linear regression; $p = 0.095$).

**Viral load in the presence of DRAMs**. DRAMs can affect both HIV-1 replicative and transmission fitness[29,30,35–37]. Therefore, we investigated the effect of common DRAMs on viral load. ART-naïve individuals with HIV diagnosed at stage 1 with L90M or K103N viruses had similar SPVL to individuals with wild-type virus. In contrast, individuals with HIV encoding a nucleoside reverse transcriptase inhibitor (NRTI) mutation with negative transmission fitness effects[29] had significantly lower SPVL (NRTI median 11,436 copies/ml versus wild-type median 18,700 copies/ml; linear regression; $p < 0.001$). This decrease in SPVL of 0.21 log$_{10}$ copies/ml in individuals with these NRTI mutations likely contributes to their observed lower transmission fitness relative to individuals with wild-type virus.

Remarkably, we observed the same relationship between clustering and viral load in ART-naïve individuals with DRAMs as was observed in individuals with wild-type virus (Table 3). The strongest association was observed in individuals with L90M-encoding virus, where we detected a 0.367 difference in the median log$_{10}$ SPVL in clustered versus nonclustered individuals (linear regression; $p = 0.003$). For individuals with K103N or NRTI DRAMs, the magnitude of this association was similar to that detected in wild-type virus, though this association did not reach statistical significance for individuals with NRTI DRAMs (linear regression; $p = 0.075$). We did not detect evidence for an interaction between clustering and these DRAMs on SPVL, even in L90M where the difference in SPVL between clustered and nonclustered individuals was three times greater than observed with wild-type virus. The absence of a significant interaction may be due to lack of power, as there were only 80 people with HIV diagnosed at stage 1 encoding L90M included in our analysis.

**Permutation analysis**. Network-based outcomes (e.g., clustering, degree centrality, and cluster size) are nonindependent and, thus, violate a fundamental assumption of the regression techniques implemented here. Therefore, we performed a network permutation analysis to assess the relationship between viral load and clustering in people with wild-type virus. We found no evidence to suggest that network structure was biasing these regression analyses. For infections diagnosed at stages 1, 2, or 3, the observed median viral loads in clustered individuals was greater than that in nonclustered individuals (stage 1, $p \leq 0.0001$; stage 2, $p \leq 0.0001$; stage 3, $p = 0.0004$; Supplementary Fig. 3). For infections

at stage 0 infection, as in the regression analysis, there was no difference in observed and permuted viral loads ($p = 0.4849$).

**Sensitivity analyses**. The relationship between viral load and the molecular transmission network in people infected with wild-type virus was robust. More conservative genetic distance thresholds are more likely to identify more recent and direct transmission partners[38,39]. Nonetheless, our findings were unaffected by using more conservative genetic distance thresholds of 0.01 and 0.005 substitutions/site to construct the molecular transmission network (Fig. 2a). Similarly, excluding the 1547 people who reported injection drug use did not affect our results (Supplementary Fig. 4), possibly owing to their rarity as well as sexual transmission of HIV among people who inject drugs in parts of the United States[40,41]. Restricting our analyses to only the 23,997 ART-naïve individuals with HIV diagnosed since 2011, thereby excluding years in which reporting was lower, did not produce meaningfully different results (Supplementary Fig. 4). Furthermore, varying the timing of first viral load measurement (i.e., first reported viral load; viral load closest to date of genotyping; and viral load closest to, but not after, date of genotyping) did not affect our results (Supplementary Fig. 4). Finally, univariate regression analyses were broadly consistent with these findings (Supplementary Table 1), though the relationship between clustering and first viral load in infections during stage 3 dissipated.

**Discussion**

Using a comprehensive molecular epidemiological approach, we investigated whether circulating wild-type subtype B strains of HIV in the United States differ in their transmission fitness and if frequently transmitted viruses in genetic clusters are more infectious than less frequently transmitted, nonclustered viruses. We found that that frequently transmitted viruses identified in genetic clusters in the inferred United States NHSS HIV-1 molecular transmission network are associated with higher viral load. Elevated viral loads associated with inferred higher transmission frequency are consistently seen across stages of HIV infection at diagnosis and in both wild-type and drug-resistant viruses. We also note that this effect was progressive with increased network connectivity, whereby higher viral loads were detected in individuals with a greater number of genetically linked partners and in larger clusters. We also observed a concomitant increase in viral load at HIV diagnosis over the past decade. Thus, these findings provide strong evidence of higher infectiousness in HIV strains frequently transmitted across the molecular transmission network. We conclude that circulating HIV subtype B strains in the United States are under natural selection to become more infectious.

Our findings also suggest an evolutionary trajectory toward higher HIV virulence, as higher viral loads increase the rate of

disease progression[21,22]. Individuals with higher viral loads are more infectious but have less time to transmit before AIDS and death, compared with individuals with lower viral loads who are less infectious but who will have more time to transmit before death. Fraser et al. hypothesized that in the absence of ART, selection would favor viruses that establish an intermediate SPVL, which maximizes infectiousness and opportunity for transmission[1,2]. For example, in Uganda, the highly virulent and infectious HIV-1 subtype D is being outcompeted by the lower virulent and less infectious HIV-1 subtype A, suggesting that the less infectious viruses that can persist longer have higher transmission fitness in that population[10,42]. In contrast, Herbeck et al. predicted that the adoption of universal test-and-treat, where all persons with an HIV diagnosis receive suppressive ART and the duration an individual can transmit virus is predominantly limited by the time between infection and diagnosis, will dampen the selective disadvantage of higher SPVL resulting in more transmissible, higher virulent HIV[43]. Our study does not find direct evidence to support the hypothesis that the current test-and-treat strategy is increasing selective pressure on HIV to evolve to be more transmissible or more virulent in the United States. Rather, this evolutionary trajectory toward higher transmissibility of HIV-1 subtype B in the United States appears unchanged during the test-and-treat era. However, we cannot exclude the possibility that our approach was not sensitive enough to detect a shift in the strength of selection.

The strength of natural selection measured in wild populations tends to be modest[44,45], with a majority of estimates of differential reproductive success (i.e., selective coefficients) <15%. Studies of selective coefficients for in vivo HIV mutations suggest that most adaptations increase replication by only 0.5–2.0%, though some mutations have larger effects[46–48]. The magnitude of differential reproductive success estimated here is in line with these expectations. Specifically, a 0.11 $\log_{10}$ copies/ml median increase in viral loads among clustered wild-type infections may be modest. Nonetheless, a viral load increase of 0.3 $\log_{10}$ copies/ml has previously been shown to increase HIV transmission by 20%[20]. Hence, the eventual impact of this selection at the population level may be important, as evidence by a 0.2 $\log_{10}$ copies/ml increase in viral load at diagnosis between 2007 and 2016. We note that this rate of increase in SPVL of 0.016 $\log_{10}$ copies/ml per year reported here is remarkably consistent with a previous meta-analysis that reported an increase in SPVL of 0.013 $\log_{10}$ copies/ml per year between 1984 and 2010[49]. This consistency suggests that a change in HIV transmissibility and virulence is not a recent phenomenon.

Our finding of progressively higher viral loads in larger clusters is important as individuals in an HIV-1 molecular transmission network whose cluster or are in disproportionately growing clusters represent priority populations for public health intervention to interrupt transmission[39,50–52]. Thus, public health interventions informed by molecular epidemiology that prioritize rapidly growing clusters with higher transmission rates may have the added benefit of prioritizing individuals with higher viral loads. Therefore, molecular epidemiological-initiated response to growing clusters could counteract the selection and propagation of more transmissible and virulent HIV, supporting further the implementation of these interventions.

Genetic clustering approaches are subject to well-characterized biases[53,54]. Extensive over-sampling of particular subpopulations or risk groups can be misinterpreted as elevated transmission rates relative to under-sampled populations[53,54]. Clustering methods are also biased toward detecting clusters comprising individuals with HIV diagnosed early in infection (as reported elsewhere[32] and seen in Table 1), because individuals separated by shorter genetic distances likely have experienced less time

since the transmission event. In fact, previous characterizations of viral load in HIV-1 molecular transmission networks have also detected a small increase in viral load in clustered individuals[19,28,55–57]; however, the confounding effects of over-sampled subpopulations, ART exposure, and time since diagnosis has previously precluded robust inference about the relationship between viral load and transmission fitness. This study was designed specifically to control for these biases. We adjusted for potential confounders like demographic and risk factor data, stratified the analyses by stage of infection at diagnosis, and explored progressive effects across the network. For individuals with HIV diagnosed at stage 0 (i.e., acute/early infection), these biases would inflate the viral load estimates for clustered individuals; however, outside of stage 0 infections, these effects would bias our results toward the null expectation. Individuals with HIV diagnosed later in infection, who are far more numerous than stage 0 cases, have higher viral loads and are less likely to cluster. Importantly, the biases discussed here would not propagate their effect across the network, and we consistently found evidence for a progressive association between viral load and network connectivity. As our previous work estimating the fitness cost of DRAMs demonstrated, despite the inherent shortcomings of molecular transmission network analysis, molecular network analysis can be an effective tool for identifying viral characteristics associated with increased transmission fitness[29,30].

Another potential source of bias concerns assigning stage of infection at diagnosis using CD4$^+$ count at diagnosis, which can be complicated by the observation that individuals with higher viral loads can experience rapid CD4$^+$ decline[58]. Hence, some individuals who had only recently been infected with HIV would be categorized as having stage 2 infection, rather than stage 1. This misclassification would again bias our results toward the null expectation, because viral load for individuals diagnosed with HIV at stage 2 have higher viral load than individuals diagnosed with HIV at stage 1.

We acknowledge that we did not have access to data on coinfection with sexually transmitted pathogens, which have previously been shown to be associated with genetic clustering[19], viral load[59–61], and HIV transmissibility[62,63]. Coinfection status could act as a confounder in our primary statistical analysis (Table 2; Fig. 1). However, like the other discussed potential sources of bias, one would not expect the effect of these coinfections on viral load to propagate across the network (as seen in Fig. 2b and Supplementary Fig. 2). For example, although infection with hepatitis C virus is predictive of HIV transmission risk[64,65] (and vice versa[66–69]), there is little overlap in path or timing of their transmission histories[70].

In conclusion, we analyzed a large HIV surveillance database from the United States and showed that subtype B HIV-1 strains have evolved to be more transmissible and virulent. Nonetheless, public health interventions that identify rapidly growing clusters and interrupt their growth—as advocated under the current *Ending the HIV Epidemic* initiative[71]—may have the added benefit of slowing HIV evolution toward higher transmissibility.

## Methods

**Study population.** The NHSS database comprised 251,754 individuals with an HIV-1 *pol* (protease and partial reverse transcriptase) sequence reported to Centers for Disease Control and Prevention (CDC) as of December 2016. We restricted our analysis to the 41,409 individuals who were documented to be ART-naïve at HIV diagnosis, had a reported HIV-1 resistance genotype (≥500 nucleotides) performed within three months of diagnosis, virus identified as subtype B, and did not report perinatal HIV exposure. We restricted our population to ART-naïve individuals, because viral load in ART-experienced individuals will reflect drug adherence rather than viral genetic underpinnings. These infections were diagnosed during 1999–2016, and 96% of which occurred after 2006, when collection of molecular data through HIV surveillance accelerated in the United States (Supplementary Fig. 1). The NHSS database also includes epidemiologic and laboratory data,

including date of diagnosis, CD4+ count and viral load results, reported transmission risk factor, and demographic data (e.g., birth sex, age at diagnosis, and race/ethnicity).

We considered only reported viral load measurements taken prior to or up to one-month post genotyping, a proxy for viral load at the time of diagnosis. For each individual, we determined the earliest viral load measurement sampled within this time frame. For infections diagnosed during stage 1, this viral load was used as a proxy for SPVL. Sensitivity analysis was also performed using the viral load measurement closest to the date of genotyping and the viral load measurement closest to, but prior to date of genotyping. Viral load measurements <200 copies/ml (indicating viral suppression) or ≥1 million copies/ml (above the limit of viral quantification assay precision) were excluded from our analysis.

HIV-1 subtype B *pol* sequences were identified using COMET[72]. We characterized the 108 DRAMs from the CDC surveillance drug resistance mutation list[73] using Sierra[74]. Stage of infection at diagnosis was determined using the US HIV surveillance case definition[75]. Stage 0 corresponds to early or acute infection recognized by a negative HIV test within six months of HIV diagnosis[33]. Stage 1 is defined by CD4+ T-cell count ≥500 cells/mm³ of blood; stage 2 is defined by CD4+ count between 200 and 499 cells/mm³; and stage 3 is defined by CD4+ <200 cells/mm³ or an AIDS-defining illness. Stage 0 classification superseded CD4+-based classifications.

This study constitutes analysis of HIV public health surveillance data and is not considered human subjects research.

**Molecular transmission network analysis**. We used HIV-TRACE to construct a molecular transmission network[31]. We selected the earliest *pol* sequence for each individual and aligned these sequences to the HXB2 *pol* reference sequence (positions 2253–3749), calculated pairwise TN93 genetic distance[76] among all pairs of sequences, and assembled transmission clusters by connecting pairs of sequences ≤0.015 substitutions/site diverged (using a nucleotide ambiguity fraction[31] of 1.5%). Individuals who were linked to ≥1 other individual were determined to be clustered in the network. This approach has previously been used for analyses of HIV surveillance data in the United States[27,29,77,78]. Sequences that were highly similar (≤0.015 substitutions/site) to the HXB2 reference sequence were filtered from the database prior to analysis.

We also constructed molecular transmission networks using more conservative genetic distance thresholds (i.e., 0.01 and 0.005 substitutions/site) and performed additional analyses excluding people who reported injection drug use.

**Regression analyses**. We investigated the relationship between viral load and clustering in the molecular transmission network using a multivariate regression analysis framework. Birth sex, transmission risk factor, race/ethnicity, age at diagnosis, year of diagnosis, and first recorded CD4+ count after diagnosis were included as covariates. Regression analyses were stratified by CD4+ stage at diagnosis[75], excluding individuals with an unknown stage of infection at diagnosis. To ensure that any inferred association between clustering and viral load was not confounded by transmitted drug resistance, we considered only individuals in the transmission network whose earliest genotype was wild-type without DRAMs. Therefore, to be clustered in the network meant that both the individual and at least one genetically linked partner had wild-type sequences.

**DRAMs**. We also investigated the association between viral load and clustering in individuals with transmitted drug resistance. We explored this association in ART-naïve individuals with HIV diagnosed during stage 1 infection with L90M (*n* = 80), K103N (*n* = 674), and the NRTI mutations which we previously documented had negative effects on transmission fitness (*n* = 260; T69N, D67N, M184V, K219Q, T69A, E44D, A62V, T69D, K70R, T215Y, K219E, K219R, T215I, F77L, D67G, and M184I)[29]. Clustering in these analyses meant that both the individual and at least one genetically linked partner shared the same DRAM.

**Molecular transmission network permutations**. We assessed the relationship between viral load and clustering through 10,000 random permutations of viral loads across the molecular transmission network. For each permuted network, we calculated the ratio of median viral loads in clustered and nonclustered individuals with wild-type virus, stratified by stage of infection at diagnosis. These permutations were used to generate a null expectation against which we compared these ratios from the observed network.

**Disclaimer**. The findings and conclusions of this report are those of the authors and do not necessarily represent the official position of the Centers for Disease Control and Prevention.

**Reporting summary**. Further information on research design is available in the Nature Research Reporting Summary linked to this article.

## Data availability

The data analyzed in this article were collected and analyzed as part of CDC routine surveillance activities reported by 30 state and local health departments (see Supplementary Table 2 for list). This analysis was conducted by only CDC employees and contractors. CDC is not permitted to share or distribute any surveillance data due to an Assurance of Confidentiality authorized under Section 308(d) of the Public Health Service Act (USA). Therefore, these data cannot be made publicly available by the authors. Each state has primary authority for determining whether their laws and regulations permit data submission to GenBank or other open databases. State and local health departments also have ability to determine whether and when these data can be shared with other researchers, as has occurred for previous studies on HIV surveillance data[41,52,79–82].

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

## Acknowledgements

J.O.W. was funded in part by the CDC, an NIH-NIAID K01 Career Development Award (K01AI110181), and an NIH-NIAID R01 (AI135992). J.O.W. is also funded by a research grant to his institution by Gilead Sciences; this grant is unrelated to the work presented here. We thank Joshua Herbeck and an anonymous reviewer for their comments on this manuscript.

## Author contributions

Conceptualization: J.O.W., A.M.O., W.M.S., N.P., J.A.J, and W.H.; Data preparation: N.S.; Data analysis: J.O.W. and C.Z.; Interpretation: J.O.W., A.M.O., W.M.S., C.Z., N.P., E.C., N.S., J.A.J., and W.H.; Drafted manuscript: J.O.W., A.M.O., and W.H. All authors had access to the study data and reviewed and approved the final manuscript.

## Competing interests

The authors declare no competing interests.
