## [Peer Review File · Nature Communications]

Reviewers' comments:

Reviewer #1 (Remarks to the Author):

In this manuscript the authors show that HIV-1 virus load is associated with clustering and with connectivity in the molecular HIV-1 transmission network. The observed associations are interesting and the paper is well written.

Comments:

1) The authors interpreted these findings as providing evidence for natural selection favouring more virulent strains. This is an overinterpretation of the presented findings. It is conceptually not fully clear how the observed clustering in the molecular HIV-1 transmission network is linked to natural selection, especially as there are many well-known confounding factors that affect clustering and transmission. Of such factors only basic demographics and stage of infection are taken into account.

2) The observed association is consistent with previously published results (Fraser et al, PNAS 2007; Fisher et al. AIDS 2010; and others). In particular, Fisher et al., present a very similar analysis, which is even adjusted for STD, but do not interpret their findings as providing strong evidence for natural selection (rather Fisher et al. interpret their findings with the appropriate caution: "We have shown an association between viral load and onward transmission, consistent with that expected by biological plausibility and that seen in heterosexual serodiscordant couple studies"). This study reduces the novelty of the presented work and should be cited.

3) Why were ART experienced individuals not included in the analysis?

Reviewer #2 (Remarks to the Author):

This is a good, robust analysis with a very interesting result.

Comments

1.a. Define “under natural selection” precisely in the paper. This will help the readers to better understand what exactly is being described... i.e. what is natural selection (a process? A pattern?), what does it result in (adaptation?), how is time (a temporal scale) involved in the definition and in the described data and analysis? Can a simple association between viral load and clustering being described as a sign of natural selection, rather than just an association between viral load and transmission? Do the authors think those things are different? The authors are assuming that viral load, because it has been shown to be heritable, can experience natural selection, and the transmissibility (clustering) is the signal of differential fitness/reproductive success. Perhaps they should spell this out directly. If this natural selection is ongoing, shouldn't we assume that adaptation is happening and that VL should be higher in later time periods (see next point)?

1.b. The main thing that I would like to see is an analysis of the time element. If this result is indeed due to ART rollout in the US, then perhaps the authors could stratify their analysis by year and see if this result does appear after ART rollout. This time-based analysis would include a test of VL (stratified, as the author's do, by stage of infection) over time, without the clustering data (what is the temporal trend in VL in this population?), and then also a redo of the overall analysis but for time slices.

2. “This finding suggests that circulating HIV strains in the United States are under natural selection favoring higher infectiousness.” At first read this sounds to be a reasonable inference from the data, but it does actually differ from previous hypotheses about HIV viral load being under natural selection for a viral load with optimal transmission, i.e. Fraser 2007 and the intermediate set point viral load (~4.5) that optimizes transmission. It is good that the authors specifically discuss this difference, but it is not until the Discussion that we get this... it might be better to briefly mention this difference earlier in the manuscript.

3. “Clustering in a molecular transmission network is an indicator of an increased transmission rate in the population.” This statement needs to be adjusted to incorporate the other known predictors/causes of clustering, as described by several previous papers (e.g. Volz, 2012; Poon, 2016). And also to reflect the fact that clustering functionally is an indicator of variation in transmission rates among subgroups in a population, with the assumption that, after adjustment for all other potential factors that affect variation in clustering rates, a subgroup with higher transmission rate will cluster more, on average, than a subgroup with lower transmission rate. (The authors do a good job of discussing all of these issues in the paper; it is just the odd single sentence like the one quoted above that may need to be revised.)

Thank you for considering our manuscript “Natural Selection Favoring More Transmissible HIV Detected in U.S. Molecular Transmission Network”. We appreciate the concerns raised by the Reviewers and have revised our manuscript to address them. Specifically, we have performed an additional series of temporal analyses suggested by Reviewer 2. These analyses, which are summarized in a newly included Figure 3, clearly demonstrate an increase in viral load over the period studied. In addition, the temporal analysis did not support the hypothesis that the natural selection for higher viral load is being driven by test-and-treat. Therefore, we have removed reference to this hypothesis from the Abstract and Introduction. In addition, we have reframed the Introduction to better introduce natural selection as it relates to HVI and viral load. And we expanded the Discussion to better frame the concept of natural selection in our study. These and other specific responses to the Reviewers comments are highlighted in BOLD below.

Sincerely,
Joel O. Wertheim

Reviewers' comments:

Reviewer #1 (Remarks to the Author):

In this manuscript the authors show that HIV-1 virus load is associated with clustering and with connectivity in the molecular HIV-1 transmission network. The observed associations are interesting and the paper is well written.

Comments:

1) The authors interpreted these findings as providing evidence for natural selection favouring more virulent strains. This is an overinterpretation of the presented findings. It is conceptually not fully clear how the observed clustering in the molecular HIV-1 transmission network is linked to natural selection, especially as there are many well-known confounding factors that affect clustering and transmission. Of such factors only basic demographics and stage of infection are taken into account.

We agree with the Reviewer that there are many factors that can influence the frequency of clustering in a population. Our analysis takes into account multiple demographic features including sex, race/ethnicity, age at diagnosis. Further, we account for reported transmission risk, which is the predominant covariate associated with clustering frequency in analyses of transmission clusters in the United States. We acknowledge that other individual-level risk factors, particularly co-infection with sexually transmitted infections (as were included in the Fisher 2010 study), are also associated with clustering in genetic analyses and can influence transmission frequency. Data on STI co-infection were not available in this analysis. We now explicitly acknowledge this limitation in the manuscript: “we did not have data on co-infection with sexually transmitted pathogens, which have previously been shown to be associated with genetic clustering”. However, we also note that STIs tend to transmit on a different time-scale than HIV. That is to say their transmission networks do not match. Therefore, one would not expect including STIs as a covariate to propagate their effect across the transmission network, as we report in Figure 2.

2) The observed association is consistent with previously published results (Fraser et al, PNAS 2007; Fisher et al. AIDS 2010; and others). In particular, Fisher et al., present a very similar analysis, which is even adjusted for STD, but do not interpret their findings as providing strong evidence for natural selection (rather Fisher et al. interpret their findings with the appropriate caution: “We have shown an association between viral load and onward transmission, consistent with that expected by biological plausibility and that seen in heterosexual serodiscordant couple studies”). This study reduces the novelty of the presented work and should be cited.

We thank the Reviewer for suggesting the inclusion of Fisher (2010) in our references. This study by Fisher represents an important step in the understanding of how viral load impacts transmission in a genetic framework and helped motivated treatment-as-prevention. Nonetheless, we do not agree with the Reviewer that the work by Fisher (2010) or by Fraser (2007) diminishes the novelty of our findings. Fraser (2007) put forth the conceptual framework for our present study, and we make every point to acknowledge how our study is indebted to this framework. In contrast, Fisher (2010) asked and answered a fundamentally different question than us. Their reporting that they “have shown an association between viral load and onward transmission, consistent with that expected by biological plausibility and that seen in heterosexual serodiscordant couple studies”. This finding can be interpreted as an effect of ART at reducing the risk of onward transmission. Our study is concerned how the viral load that stems from additive genetic variation in the virus (i.e., heritable variation) influences onward transmission.

3) Why were ART experienced individuals not included in the analysis?

The heritability of set-point viral load (and the correlated viral loads at other stages of infection) is only measured and valid in ART-naïve persons. The purpose of this study was to understand how the inherent viral load (e.g., set-point viral load for individuals diagnosed with Stage 1 infection) influences HIV transmission. These viral loads can only be estimated in ART-naïve individuals. Once an individual is ART-experienced, their viral load has far more to do with drug adherence than the underlying genetics of the virus. Measuring viral fitness would be exceedingly difficult, if not impossible, in this context. Restricting our analysis to ART-naïve individuals, we are answering a different question that previous studies (like Fisher 2010, Poon 2015, Brenner 2017, etc.).

We have updated the methods to reflect this reasoning, stating “We restricted our population to ART-naïve individuals, because viral load in ART-experienced individuals will reflect drug adherence rather than viral genetic underpinnings”.

Reviewer #2 (Remarks to the Author):
Josh Herbeck

This is a good, robust analysis with a very interesting result.

Comments

1.a. Define “under natural selection” precisely in the paper. This will help the readers to better understand what exactly is being described... i.e. what is natural selection (a process? A pattern?), what does it result in (adaptation?), how is time (a temporal scale) involved in the definition and in the described data and analysis? Can a simple association between viral load and clustering being described as a sign of natural selection, rather than just an association between viral load and transmission? Do the authors think those things are different? The authors are assuming that viral load, because it has been shown to be heritable, can experience natural selection, and the transmissibility (clustering) is the signal of differential fitness/reproductive success. Perhaps they should spell this out directly. If this natural selection is ongoing, shouldn't we assume that adaptation is happening and that VL should be higher in later time periods (see next point)?

We agree with the Reviewer on this suggestion. We have restructured the Introduction to better frame our study in this context. The first sentences of the manuscript now read: “Natural selection is the process by which the differential reproductive success of an organism with particular trait, whose variance in the population has a genetic underpinning (i.e., is heritable), leads to change in a population. In human immunodeficiency virus (HIV), a trait that is likely shaped by natural selection is viral load, the concentration of HIV in blood”. Further, we introduce how set-point viral load and its heritability fit into this concept.

The second paragraph of the Introduction now focuses on the complications of measuring natural selection as it relates to viral load “Measuring natural selection associated with SPVL is not straightforward, because higher viral loads are also associated with higher infectiousness”. This paragraph then leads into how genetic transmission networks can be used to overcome this issue. We thank the Reviewer for calling our attention to this issue. We believe that a slightly modified, but substantially restructured, Introduction will better make a case for our approach and provides improved contextualization.

Further, we have expanded on this concept in the Discussion, framing our results in the context of other studies estimating the strength of natural selection in HIV and other organisms:

“The strength of natural selection measured in wild populations tends to be modest, with a majority of estimates of differential reproductive success (i.e., selective coefficients) less than 15%. Studies of selective coefficients for in vivo HIV mutations suggest that most adaptations increase replication by only 0.5 to 2.0%, though some mutations have large effects. Therefore, the magnitude differential reproductive success we estimated here is in line with expectation. Specifically, a 0.11 log₁₀ copies/ml median increase in viral loads among clustered wildtype infections may be modest. However, its eventual impact at the population level may be important, as evidence by a 0.2 log₁₀ copies/ml increase between 2007 and 2016. For example, a viral load increase of 0.3 log₁₀ copies/ml has previously been shown to increase HIV transmission by 20%.”

1.b. The main thing that I would like to see is an analysis of the time element. If this result is indeed due to ART rollout in the US, then perhaps the authors could stratify their analysis by year and see if this result does appear after ART rollout. This time-based analysis would include a test of VL (stratified, as the author's do, by stage of infection)

over time, without the clustering data (what is the temporal trend in VL in this population?), and then also a redo of the overall analysis but for time slices.

We thank the Reviewer for their thoughtful suggestion. We have updated the manuscript to include the suggested temporal analysis. First, we report the relationship between year of diagnosis and log viral load in both univariate and multivariate regression models. These results are presented in a new Figure (Figure 3). We report significant positive associations between viral load and time for individuals diagnosed with Stage 1, 2, and 3 infections. These findings support the hypothesis that HIV subtype B in the United States is evolving to be more transmissible and virulent over the past decade. For this analysis, we applied the same inclusion criteria to these analyses as we did for the primary analyses. The reason for this decision is that we were concerned with viral load in ART-naïve individuals with wild-type Subtype B virus. We only know the subtype and DRAM content of viruses with a reported sequence.

With regards to the Reviewer's second suggestion, we opted to perform statistical tests for a change in the relationship to between viral load and clustering over time. We do not report specific year-long time-slices, because of the loss of statistical power that came from dividing our analysis into tenths. Rather, we report the results of interaction models in the multivariate regression framework. This approach should be more powered and provides a direct statistical test of the hypothesis that the association between viral load and clustering has increased over time. In short, we do not find statistical support for the hypothesis that test-and-treat has increased selection for more transmissible, higher viral load HIV.

We have updated the manuscript to reflect these findings: "Our study does not find direct evidence to support the hypothesis that the current test-and-treat strategy is placing selective pressure on HIV to evolve to be more transmissible or more virulent in the United States. Rather, this evolutionary trajectory towards higher transmissibility of HIV-1 subtype B in the United States appears unchanged during the test-and-treat era. However, we cannot exclude the possibility that our approach was not sensitive enough to detect a shift in the strength of selection due to test-and-treat".

We have also removed reference to this hypothesis in the Abstract and the Introduction, as it is no longer supported by our results.

As an aside, we appreciate that the Reviewer would push for a deeper inquiry into findings that initially appeared to confirm their own previously published study.

2. "This finding suggests that circulating HIV strains in the United States are under natural selection favoring higher infectiousness." At first read this sounds to be a reasonable inference from the data, but it does actually differ from previous hypotheses about HIV viral load being under natural selection for a viral load with optimal transmission, i.e. Fraser 2007 and the intermediate set point viral load (~4.5) that optimizes transmission. It is good that the authors specifically discuss this difference, but it is not until the Discussion that we get this... it might be better to briefly mention this difference earlier in the manuscript.

We agree with the Reviewer and now make it explicitly clear how this process would differ from the scenario put forth by Fraser. This text can be found in the new introductory paragraph: "In the context of HIV and set-point viral load, natural selection will favor HIV with a higher set-point viral load if that higher set-point viral load increases the number of subsequent transmission events over the duration of viral infection. This process would result in a population of viruses whose set-point viral has increased transmissibility".

3. "Clustering in a molecular transmission network is an indicator of an increased transmission rate in the population." This statement needs to be adjusted to incorporate the other known predictors/causes of clustering, as described by several previous papers (e.g. Volz, 2012; Poon, 2016). And also to reflect the fact that clustering functionally is an indicator of variation in transmission rates among subgroups in a population, with the assumption that, after adjustment for all other potential factors that affect variation in clustering rates, a subgroup with higher transmission rate will cluster more, on average, than a subgroup with lower transmission rate. (The authors do a good job of discussing all of these issues in the paper; it is just the odd single sentence like the one quoted above that may need to be revised.)

We agree with reviewer that this sentence, which was situated in the Results section lacked the appropriate context and nuance. We have updated the manuscript to simply state our findings: "We detected a robust association between viral load and clustering in the inferred molecular transmission network".

REVIEWERS' COMMENTS:

Reviewer #1 (Remarks to the Author):

I would like to thank the authors for responding to my concerns.

I am however not fully convinced by their response to my first point. I do not think that this concern invalidates their very interesting analysis, but it should be spelled out more clearly as a limitation that clustering is only a very indirect way to measure viral fitness. Concerning, STD (which are only one reason why the link between clustering and fitness is expected to be weak): My concern is that STD can act as a classical confounder by increasing both virus load ([1-4]) and the transmission risk (via virus-load independent pathways; e.g. ulcers [5]). Similarly, the confounder could be behaviour (e.g. number of partners, condoms) causing more HIV transmission but also more STDs, which in turn increase virus load. In both cases there would be an association between virus load and transmission (and therefore clustering) without “additive genetic variation in the virus (i.e., heritable variation) influencing onward transmission.” I cannot see how the different time-scales mentioned by the authors would reduce this confounding.

1) N. Nagot, A. Ouédraogo, V. Foulongne et al., “Reduction of HIV-1 RNA levels with therapy to suppress herpes simplex virus,” *The New England Journal of Medicine*, vol. 356, no. 8, pp. 790–799, 2007

2) R. Palacios, F. Jiménez-Oñate, M. Aguilar et al., “Impact of syphilis infection on HIV viral load and CD4 cell counts in HIV-infected patients,” *Journal of Acquired Immune Deficiency Syndromes*, vol. 44, no. 3, pp. 356–359, 2007

3) T. Schacker, J. Zeh, H. Hu, M. Shaughnessy, and L. Corey, “Changes in plasma human immunodeficiency virus type 1 RNA associated with herpes simplex virus reactivation and suppression,” *Journal of Infectious Diseases*, vol. 186, no. 12, pp. 1718–1725, 2002.

4) D. Serwadda, R. H. Gray, N. K. Sewankambo et al., “Human immunodeficiency virus acquisition associated with genital ulcer disease and herpes simplex virus type 2 infection: a nested case-control study in Rakai, Uganda,” *Journal of Infectious Diseases*, vol. 188, no. 10, pp. 1492–1497, 2003.

5) Ward H, Rönn M.; Contribution of sexually transmitted infections to the sexual transmission of HIV. *Curr Opin HIV AIDS*. 2010 Jul;5(4):305-10.

Reviewer #2 (Remarks to the Author):

The authors have responded thoroughly to my previous comments and suggestions. I would support acceptance and publication.

I only have a couple new comments.

1. I know that this is probably a bit pedantic and nit-picky, and that the authors fully know this (and address it well in the manuscript in a different spot), but I have a small problem with the following statement in the Abstract:

"We inferred HIV genetic transmission clusters using data from the United States National HIV Surveillance System and found that **frequently transmitted viruses in clusters had higher viral load at diagnosis**."

Technically we don't know if this is true, as we are just making the assumption that clustered viruses were more frequently transmitted than unclustered viruses. i.e. We don't actually know the frequency of transmission for clustered vs unclustered viruses.

This assumption is also unstated in this sentence from the Discussion:

"We show that the higher viral loads in frequently transmitted viruses are ..."

Nit-picky, I know, and I apologize.

2. In the "Viral load over time" section of the Results, which I am very happy to see added to the analysis and manuscript, it took me a second read to figure out whether this analysis was looking at just clustered individuals or all individuals. It might help to make this more clear straight away--this is essentially an epi analysis much like the many SPVL ~ time analyses that have been published, as it does not include sequence data.

2.a. It is interesting that the addendum to the "Viral load over time section" states that this SPVL/VL trend is similar for clustered and unclustered individuals... this could use a bit of explanation. The result of selection at transmission for higher VL viruses is not constrained to viruses that transmit (cluster): it results in new infections with higher VLs, even if those new infections do not go on to clusters/transmit themselves. Obvious, but an interesting note on the impact of adaptation in a population.

3. I hesitate to suggest this, because I try not to promote my earlier work, and god forbid I would ask, as a reviewer, that authors cite any of my papers, BUT the estimated annual changes in VL (0.016, 0.010, 0.008; for different infection stages; Table 2) are very close to the estimated annual change in SPVL that I estimated in my 2012 meta-analysis of SPVL trend studies (effect=0.013 log(10) copies/ml per year; <https://www.ncbi.nlm.nih.gov/pubmed/22089381>). Personally I think that would warrant a mention--regardless of who authored that meta-analysis--as it is surprisingly consistent!

Thank you for considering our revised manuscript “Natural Selection Favoring More Transmissible HIV Detected in U.S. Molecular Transmission Network”. We appreciate the additional feedback from the Reviewers. We have revised our manuscript to incorporate their perspectives. Our specific responses to the Reviewers comments are highlighted in BOLD below.

**Sincerely,
Joel O. Wertheim**

Reviewers' comments:

Reviewer #1 (Remarks to the Author):

I would like to thank the authors for responding to my concerns. I am however not fully convinced by their response to my first point. I do not think that this concern invalidates their very interesting analysis, but it should be spelled out more clearly as a limitation that clustering is only a very indirect way to measure viral fitness. Concerning, STD (which are only one reason why the link between clustering and fitness is expected to be weak): My concern is that STD can act as a classical confounder by increasing both virus load ([1-4]) and the transmission risk (via virus-load independent pathways; e.g. ulcers [5]). Similarly, the confounder could be behaviour (e.g. number of partners, condoms) causing more HIV transmission but also more STDs, which in turn increase virus load. In both cases there would be an association between virus load and transmission (and therefore clustering) without “additive genetic variation in the virus (i.e., heritable variation) influencing onward transmission.” I cannot see how the different time-scales mentioned by the authors would reduce this confounding.

1) N. Nagot, A. Ouédraogo, V. Foulongne et al., “Reduction of HIV-1 RNA levels with therapy to suppress herpes simplex virus,” *The New England Journal of Medicine*, vol. 356, no. 8, pp. 790–799, 2007

2) R. Palacios, F. Jiménez-Oñate, M. Aguilar et al., “Impact of syphilis infection on HIV viral load and CD4 cell counts in HIV-infected patients,” *Journal of Acquired Immune Deficiency Syndromes*, vol. 44, no. 3, pp. 356–359, 2007

3) T. Schacker, J. Zeh, H. Hu, M. Shaughnessy, and L. Corey, “Changes in plasma human immunodeficiency virus type 1 RNA associated with herpes simplex virus reactivation and suppression,” *Journal of Infectious Diseases*, vol. 186, no. 12, pp. 1718–1725, 2002.

4) D. Serwadda, R. H. Gray, N. K. Sewankambo et al., “Human immunodeficiency virus acquisition associated with genital ulcer disease and herpes simplex virus type 2 infection: a nested case-control study in Rakai, Uganda,” *Journal of Infectious Diseases*, vol. 188, no. 10, pp. 1492–1497, 2003.

5) Ward H, Rönn M.; Contribution of sexually transmitted infections to the sexual transmission of HIV. *Curr Opin HIV AIDS*. 2010 Jul;5(4):305-10.

We agree with the Reviewer’s assessment that STD co-infection could be a confounder, in the classical sense. We have reviewed and included the helpful references provided by the Reviewer to help illustrate this point in the Discussion section of the manuscript. We maintain, however, that the progressive relationship we report between the extent of cluster and viral load suggests that this relationship is indeed robust. To clarify, these co-infecting pathogens are not transmitted along the same path or at the same time as

HIV transmission. We have included the following paragraph in the Discussion section to appropriately convey these points.

“We acknowledge that we did not have access to data on co-infection with sexually transmitted pathogens, which have previously been shown to be associated with genetic clustering¹⁹, viral load⁵⁹⁻⁶¹, and HIV transmissibility^{62,63}. Co-infection status could act as a cofounder in our primary statistical analysis (Table 2; Figure 1). However, like the other discussed potential sources of bias, one would not expect the effect of these co-infections on viral load to propagate across the network (as seen in Figure 2B and Supplementary Figure 2). For example, although infection with hepatitis C virus is predictive of HIV transmission risk^{64,65} (and vice-versa⁶⁶⁻⁶⁹), there is little overlap in path or timing of their transmission histories⁷⁰.”

Reviewer #2 (Remarks to the Author):

The authors have responded thoroughly to my previous comments and suggestions. I would support acceptance and publication.

I only have a couple new comments.

1. I know that this is probably a bit pedantic and nit-picky, and that the authors fully know this (and address it well in the manuscript in a different spot), but I have a small problem with the following statement in the Abstract:

"We inferred HIV genetic transmission clusters using data from the United States National HIV Surveillance System and found that frequently transmitted viruses in clusters had higher viral load at diagnosis."

Technically we don't know if this is true, as we are just making the assumption that clustered viruses were more frequently transmitted than unclustered viruses. i.e. We don't actually know the frequency of transmission for clustered vs unclustered viruses.

We thank the reviewer for their careful attention to detail. We agree with their sentiment and have modified the sentence to read “Here, we reconstruct HIV genetic transmission clusters using data from the United States National HIV Surveillance System and report that viruses in clusters, inferred to be frequently transmitted, have higher viral loads at diagnosis.”

This assumption is also unstated in this sentence from the Discussion:

"We show that the higher viral loads in frequently transmitted viruses are ..."

Nit-picky, I know, and I apologize.

We agree that this sentence also glossed-over this nuance. We have replaced it with: “. Viruses with inferred higher transmission frequency show consistently higher viral loads across stages of HIV infection and with both wildtype and drug-resistant viruses.”

2. In the "Viral load over time" section of the Results, which I am very happy to see added to the analysis and manuscript, it took me a second read to figure out whether this analysis was looking at just clustered individuals or all individuals. It might help to make this more clear straight away--this is essentially an epi analysis much like the many SPVL ~ time analyses that have been published, as it does not include sequence data.

We thank the reviewer for pointing this source of ambiguity. The figure legend has now been updated to include the following text: "Figure 3. Viral load at diagnosis over time. These plots include the first viral load measurement for individuals with (A) Stage 0, (B) Stage 1, (C) Stage 2, and (D) Stage 3 infection. Plots included both clustered and non-clustered individuals, all of whom were antiretroviral therapy (ART)-naïve with a reported subtype B genotype and no evidence of drug resistance associated mutations (DRAMs). Solid black lines indicate slope (β) from univariate regression analysis comparing \log_{10} viral load (copies/ml) and year of diagnosis. For display purposes, viral loads for individuals with a diagnosis before 2005 are omitted, and the viral load results are plotted against their date of diagnosis, rather than only year."

2.a. It is interesting that the addendum to the "Viral load over time section" states that this SPVL/VL trend is similar for clustered and unclustered individuals... this could use a bit of explanation. The result of selection at transmission for higher VL viruses is not constrained to viruses that transmit (cluster): it results in new infections with higher VLs, even if those new infections do not go on to clusters/transmit themselves. Obvious, but an interesting note on the impact of adaptation in a population.

We agree that this finding, which was in contrast to our initial expectation, could benefit from additional explanation in the text of the manuscript. The use of clustered/non-clustered dichotomy is meant to divide the population into viruses that transmit frequently versus those that do not. If, as our results now suggest, that this trend in increasing viral load has been occurring for many, many years (see response below), then non-clustered viruses do not represent viruses that do not transmit. Rather, they represent viruses that, on average, do not transmit as frequently. The reviewer is correct that this change in viral load is affecting the entire population, rather than only clustered viruses.

We have added the following clarification sentence to that paragraph: "If natural selection is acting to increase viral load, the strength of this selection has not changed over the time-period analyzed. Moreover, this increase in viral load over time is occurring across the entire sampled HIV population, rather than only in clustered viruses."

We have also revised the last paragraph of the Discussion, which may be contributed to this confusion.

3. I hesitate to suggest this, because I try not to promote my earlier work, and god forbid I would ask, as a reviewer, that authors cite any of my papers, BUT the estimated annual changes in VL (0.016, 0.010, 0.008; for different infection stages; Table 2) are very close to the estimated annual change in SPVL that I estimated in my 2012 meta-analysis of SPVL trend studies (effect=0.013 \log_{10} copies/ml per year; <https://www.ncbi.nlm.nih.gov/pubmed/22089381>). Personally I think that would warrant a mention--regardless of who authored that meta-analysis--as it is surprisingly consistent!

We agree that this remarkable consistency deserves an explicit discussion. We are aware of this work and regret not including it as a reference in the initial revised manuscript. We have added this reference to the Discussion Section:

“Hence, the eventual impact of this selection at the population level may be important, as evidence by a 0.2 log₁₀ copies/ml increase in viral load at diagnosis between 2007 and 2016. We note that this rate of increase in SPVL of 0.016 log₁₀ copies/ml per year reported here is remarkably consistent with a previous meta-analysis that reported an increase in SPVL of 0.013 log₁₀ copies/ml per year between 1984 and 2010⁴⁹. This consistency suggest that a change in HIV transmissibility and virulence is not a recent phenomenon.”